# Detrimental Effect of Ozone on Pathogenic Bacteria

**DOI:** 10.3390/microorganisms10010040

**Published:** 2021-12-26

**Authors:** Karyne Rangel, Fellipe O. Cabral, Guilherme C. Lechuga, João P. R. S. Carvalho, Maria H. S. Villas-Bôas, Victor Midlej, Salvatore G. De-Simone

**Affiliations:** 1FIOCRUZ, Center for Technological Development in Health (C.D.T.S.), National Institute of Science and Technology for Innovation in Neglected Population Diseases (INCT-IDPN), Rio de Janeiro 21040-900, RJ, Brazil; guilherme.lechuga@cdts.fiocruz.br (G.C.L.); joaoprsc@id.uff.br (J.P.R.S.C.); 2Microbiology Department, National Institute for Quality Control in Health (I.N.C.Q.S.), FIOCRUZ, Rio de Janeiro 21040-900, RJ, Brazil; fellipe.cabral@incqs.fiocruz.br (F.O.C.); maria.villas@incqs.fiocruz.br (M.H.S.V.-B.); 3Department of Molecular and Cellular Biology, Biology Institute, Federal Fluminense University, Niteroi 22040-036, RJ, Brazil; 4Laboratory of Cellular and Ultrastructure, Oswaldo Cruz Institute, FIOCRUZ, Rio de Janeiro 21040-900, RJ, Brazil; victor.midlej@ioc.fiocruz.br; 5Epidemiology and Molecular Systematic Laboratory, Oswaldo Cruz Institute, FIOCRUZ, Rio de Janeiro 21040-900, RJ, Brazil

**Keywords:** ozone, detrimental effect, pathogenic bacteria, antimicrobial resistance, R.O.S., *Acinetobacter baumannii*, *Pseudomonas aeruginosa*

## Abstract

(1) Background: Disinfection of medical devices designed for clinical use associated or not with the growing area of tissue engineering is an urgent need. However, traditional disinfection methods are not always suitable for some biomaterials, especially those sensitive to chemical, thermal, or radiation. Therefore, the objective of this study was to evaluate the minimal concentration of ozone gas (O_3_) necessary to control and kill a set of sensitive or multi-resistant Gram-positive and Gram-negative bacteria. The cell viability, membrane permeability, and the levels of reactive intracellular oxygen (ROS) species were also investigated; (2) Material and Methods: Four standard strains and a clinical MDR strain were exposed to low doses of ozone at different concentrations and times. Bacterial inactivation (cultivability, membrane damage) was investigated using colony counts, resazurin as a metabolic indicator, and propidium iodide (PI). A fluorescent probe (H_2_DCFDA) was used for the ROS analyses; (3) Results: No reduction in the count colony was detected after O_3_ exposure compared to the control group. However, the cell viability of *E. coli* (30%), *P. aeruginosa* (25%), and *A. baumannii* (15%) was reduced considerably. The bacterial membrane of all strains was not affected by O_3_ but presented a significant increase of ROS in *E. coli* (90 ± 14%), *P. aeruginosa* (62.5 ± 19%), and *A. baumanni* (52.6 ± 5%); (4) Conclusion: Low doses of ozone were able to interfere in the cell viability of most strains studied, and although it does not cause damage to the bacterial membrane, increased levels of reactive ROS are responsible for causing a detrimental effect in the lipids, proteins, and DNA metabolism.

## 1. Introduction

“Deaf Endemic” is a term used by the World Health Organization (WHO) as a reference to health care-associated infections (HAIs), identified as the most frequent adverse effects during health care provision [1]. HAIs comprise any infection acquired after a patient’s admission to the hospital, which may occur during hospitalization or shortly after discharge, as long as it is related to hospitalization or the procedures performed during the period [2,3]. 

In hospitals, the surfaces, equipment, and medical devices play an essential role in spreading HAIs, as secondary reservoirs promote cross-contamination [4]. The hands of health professionals correspond to the most common means of transferring pathogens [5]. In intensive care units (ICUs), even after adopting strict cleaning and disinfection protocols, many patients are infected with HAIs [6,7,8]. These infections are more frequent in these units, where outbreaks usually originate [9] since the use of antibiotics is approximately 10 times greater than in general hospital wards [10,11]. In addition, HAIs lead to an increase in deaths (morbidity and mortality), favors the development of resistant pathogens, prolong hospital stay, and, consequently, health costs [12,13,14,15,16,17,18]. 

The National Health Security Network of the Centers for Disease Control and Prevention (CDC) estimated 687,000 HAIs in U.S. acute care hospitals, causing 72,000 deaths and estimated costs of USD 97–147 billion annually [19,20]. Furthermore, in low and middle income countries, such as Brazil, the frequency of infection acquired in the I.C.U. is at least 2–3 and up to 5 times higher than in high-income countries [21,22], and 5–10 times greater than those acquired in general clinical wards and surgery [23]. 

In Brazil, a study found an incidence of adverse events of 8.4% in hospitalized patients, with HAI being the second most frequent (20% of cases) after surgical events (24.6%) [24]. A multi-stage survey, carried out in Brazil with a team of trained data collectors, detected an overall prevalence rate of HAI of 10.8% [25]. Another study carried out in Brazil demonstrated the general prevalence of HAIs, which was higher (51.2%) than that reported for the USA (6.1%) and Europe (48.4%) in multicenter surveillance studies [26,27,28], but similar to other Brazilian studies [29,30,31]. In this study, the highest prevalence (78.6%) was observed in a unit located in the northern region of the state, the region with the lowest income [32].

Antimicrobial resistance is one of the most complex health challenges today. Despite warnings from international organizations, the world has long ignored warnings that antibiotics and other drugs are losing their effectiveness after decades of overuse and improper use in human medicine, animal health, and agriculture [33,34]. Common diseases, such as pneumonia, postoperative infections, and diarrhea, are becoming increasingly intractable due to the emergence and spread of drug resistance. Unfortunately, antibiotic consumption increases in some countries, especially in low and middle-income countries [35]. In the past few years, some high-income countries have decreased the consumption of antibiotics, suggesting that the educational/regulatory strategies developed in recent years significantly reduced consumption [36]. However, if no policy change is made, the projection of global antibiotic consumption in 2030 will be up to 200% [35]. One study compared the type of pathogens and the susceptibility of the isolated strains and the use of antibiotics in surgical departments. The study monitored all the patients admitted for one year where the total consumption of antibiotics was found to be 479.18 DDD/1000 patient-days in the surgical sections. The pattern of antibiotic use by class showed that the most frequently prescribed antibiotics in all departments of the hospital were cephalosporins (54.30%), followed by fluoroquinolones (10.99%), penicillin and beta-lactamase inhibitor mixtures (10.76%), aminoglycosides (7.65%), and carbapenems (5.46%). Linezolid, macrolides, and tetracyclines were less frequently prescribed in the whole hospital [37].

The antimicrobial resistance is one of the three most critical problems for human health [38,39], and the most common and severe multidrug-resistant pathogens (MDR) that cause HAIs are *Clostridium difficile* and the bacteria included in the acronym “E.S.K.A.P.E.” (*Enterococcus* spp., *Staphylococcus aureus*, *Klebsiella* spp., *Acinetobacter spp*., *Pseudomonas aeruginosa*, and Enterobacteriaceae) [40]. In addition, many bacteria exhibit antimicrobial resistance and can cause bloodstream infections, urinary tract, severe pneumonia, and surgical site infection [9]. Thus, the only possible defense against the threat of antimicrobial resistance and the possibility of a post-antibiotic era is all stakeholders’ global and coordinated effort to support the development of new antimicrobial drugs, diagnostics, vaccines, and other tools. The hospital environment is an essential reservoir of microorganisms, especially multidrug-resistant ones. Active screening for resistant multidrug strain carriers remains a crucial component of infection control policy in any healthcare setting. Another study analyzed the incidence and pattern of bacterial colonization including MDR. Gram-positive (methicillin-resistant *Staphylococcus aureus* (MRSA), vancomycin-resistant *Enterococcus* spp. (VRE)) and Gram-negative (extendedspectrum beta-lactamase producers (ESBL), and carbapenemase-resistant Enterobacteriaceae (CRE)) germs in ICU. Patients upon admission and after 7 days showed bacterial colonization on admission was detected in a quarter of patients, and carbapenemase-producing bacteria were the most common colonizers (21.16%). The 7-day ICU stay also proved to be an increased risk for ESBL and CRE infection [41]. 

Among the factors that favor the contamination of the health service environment, the hands of health professionals in contact with the surfaces are important to mention as well as maintenance of damp, wet and dusty surfaces, imperfect coatings, and organic matter maintenance [29,42]. The presence of dirt, especially organic matter of human origin, can serve as a substrate for the proliferation of microorganisms or favor the company of vectors, which can passively carry these agents. Hence the importance of cleaning and rapid disinfection of any area with organic matter, regardless of the hospital area [43,44,45,46]. Although effective disinfection of surfaces and the environment is considered one of the primary measures to control the spread of HAIs, hospital surfaces remain neglected reservoirs.

Some traditional disinfecting hospital environments’ apparent lack of effectiveness [47,48] stimulated the search for new decontamination methods that are also “environmentally friendly.” As a result, there has been engagement in using ozone gas as a chemical element for antimicrobial control in several areas [49,50,51,52,53,54] and as a disinfectant [55,56,57,58].

Ozone (O_3_) is a bioactive oxidizing disinfectant that decomposes into O_2_ and O_1_, and the latter molecule is highly reactive, causing the breakdown of bacterial cell walls and changing the function of proteins and carbohydrates [57,58]. It has been reported that O_3_ can be employed as a bactericidal agent under many forms, such as ozonized oil [59], ozonized water [60], ozonized saline solution [61], ozone associated with other substances [62], and more often, the gaseous O_3_/O_2_ mixture [63]. Due to this property, it has a high level of effectiveness in eliminating bacteria, fungi, and molds [64,65], and in inactivating viruses [66,67]. However, the point of ozone in treating microorganisms, especially bacteria and viruses, is related to various factors, i.e., ozone concentration, the temperature of the environment, humidity of the environment, and exposure time [68]. Gaseous and water-based O_3_, used for disinfection in the food industry and water systems, is currently being studied to interrupt biofilms in periodontics [55,56,57,58]. O_3_ has a half-life of approximately 20 min in the gaseous phase, which has restricted some applications before low concentration exposures for prolonged periods, with limited effectiveness [69]. However, O_3_ has already been applied to clean hospital clothing [70], and a recent study demonstrated the eradication of M.R.S.A. in a nurse’s home environment after O_3_ decontamination [50]. Ozone generated by ozone purifiers in low concentrations is safe when following the manufacturer’s scheme [71]. Gaseous ozone in relatively high concentrations (25 parts per million—ppm) has also been used to inactivate norovirus and bacteria in office and hotel rooms, removing this ozone after using a system purifier [51,52]. In medicine, O_3_ has already been used in different forms of application (parenteral or local), aiming to combat ischemia, joint diseases, immunosuppression, degenerative diseases, and infections [72,73,74], and for therapeutic purposes [75,76]. However, in high concentrations, it becomes toxic [77]. Thus, different beneficial effects can be obtained if used correctly and controlled.

Therefore, with the knowledge of the emergence of microorganisms resistant to conventional antimicrobials and that the use of O_3_ is considered an effective disinfectant of low cost, we have determined the minimal concentration of O_3_ necessary to control and kill a set of Gram-positive and Gram-negative bacteria. Furthermore, we have analyzed cell viability, membrane permeability, intracellular reactive oxygen species (ROS) levels, and ultrastructural bacterial membrane damage in this work. 

## 2. Materials and Methods

### 2.1. Bacterial Strains 

Standard strains (*Staphylococcus aureus* (ATCC 6538), *Salmonella enterica* subsp enterica serovar *choleraesuis* (ATCC 10708), *Escherichia coli* (ATCC 25922), and *Pseudomonas aeruginosa* (ATCC 15442) were obtained from the American Type Culture Collection (ATCC) (Plast Labor Ind Com E.H. Lab Ltd., R.J., Brazil). The MDR strain of *Acinetobacter baumannii* with origin from the local hospital, carrying the blaOXA-23 gene and representing one of the genotypes disseminated in Brazil (ST15/CC15), was also used. This strain was provided by Dr. Maria H. S. Villas-Bôas (National Institute for Quality Control in Health of the Oswaldo Cruz Foundation-INCQS/FIOCRUZ). These bacterial strains were initially cultivated according to the instructions of the ATCC, aliquoted, and stored in cryotubes containing tryptic soy broth (TSB, Difco) with 20% glycerol (v/v) and kept at −20 °C for later use. 

### 2.2. Ozone Generating and Monitoring

An air purifier/sterilizer with O_3_ (SANITECH O3 PURI-MU, Astech Serv. and Fabrication Ltd., Petropolis, Brazil) with the capacity to treat the room air up to 30 m^3^ was used. The monitoring and measurement of the environmental concentration of O3 emitted were realized using a portable gas detector (BH-90A, Forensics L.L.C., Los Angeles, CA, USA). This detector monitored combustible and toxic gases using a built-in MCU sensor controller with a range of 0 to 20 ppm with two alarms set at 5 ppm and 10 ppm. 

### 2.3. Inoculation of the Test Surface

The strains were removed from the freezer stock culture for bacterial reactivation, sown in T.S.B., and incubated at 37 °C for 24 h. After the microorganisms were suspended in sterile 0.85% saline and the concentration of 108 CFU ml^−1^ was determined with a densitometer (Densichek Plus, BioMérieux, Durham, NC, USA). The successive dilutions (10^4^, 10^3^, and 10^2^ CFU ml^−1^) were made in the brain heart infusion broth (BHI). One hundred microliter aliquots of each bacterial suspension (*S. aureus, S. enterica, E. coli, P. aeruginosa, and A. baumannii* in different concentrations (10^4^, 10^3^, and 10^2^ CFU ml^−1^) were plated in triplicate by spread plate on Triptona Soy Agar (TSA; Difco) and incubated at 37 °C for 24 h. 

### 2.4. Ozone Treatment 

The inoculated plates containing the different microorganisms were placed on a laboratory bench measuring 2.55 m^2^ (2.50 m × 0.62 m) equidistant and symmetrically opposed, then opened and exposed to only one SANITECH O3 PURI-MU device (turned on 1 h before starting the experiment and approximately 1.10 m away) for 10 h and 12 h exposure to ozone in a test room measuring about 38 m^3^ (3.60 m × 3.65 m × 2.90 m), and kept closed, except when measuring and monitoring the environmental O3 concentration. After the exposure time, the plates were closed and incubated at 37 °C for 24 h. 

As a positive control of the assay, plates with TSA containing the same bacterial suspensions were used without exposure to O_3_. These control plates remained at room temperature and were incubated at 37 °C for 24 h together with the plates exposed to O_3_. A plate containing only TSA was used as a negative control. The test was performed in triplicate. Colony counting was performed only on plates with many colonies from 0 to 300. 

### 2.5. Cell Viability

The cell viability was measured on selected bacterial suspension of 10^3^ CFU ml^−1^ after 10 h exposure to O_3_ based on previous results (cell count- CFU ml^−1^). The entire previous experiment was performed again (at the defined concentration and time), and after 24 h of incubation, three distinct colonies from each plate were inoculated separately in a test tube containing BHI broth (Difco). As a positive control of the assay, we performed the same procedure with the plates that were not exposed to O_3_, where three distinct colonies of each dish were inoculated separately in a test tube containing BHI broth (Difco). Afterward, 100 μL of the bacterial suspension of each colony was transferred, in triplicate, to the wells of the 96-well microplate, which was incubated at 37 °C for 24 h. Each strain was tested in duplicate and detected bacterial growth by adding 0.02% resazurin (7-hydroxy-phenoxazin-3-one 10-oxide; Sigma-Merck, St Louis, MO, USA) 1 h incubation [78]. Resazurin is a non-toxic, non-fluorescent blue reagent that, after enzymatic reduction, becomes highly fluorescent. This conversion occurs only in viable cells, and as such, the amount of resorufin produced is proportional to the number of viable cells in the sample [79,80,81]. As a negative control, we used BHI broth, and the measured at 590 nm was made on an ELISA plate reader (Flex Station 3, Molecular Devices, San José, CA, USA).

The collected data were analyzed using the program R (version 3.6.0) and R Studio, where the paired *t*-test was applied to compare the statistical significance between the two samples (with and without treatment with O_3_) with ≤0.01. Each experiment was repeated three times for each microorganism treated with O_3_. 

### 2.6. Live/Dead Assay

The effect of ozone on bacteria membrane permeability was measured using fluorescent probes to stain live (Syto9) and dead bacteria with the disrupted membrane (propidium iodide; PI). Briefly, bacteria suspension (10^3^ CFU ml^−1^) was cultivated in BHI broth for 24 h in 24 well plates in the presence of O_3_. After exposure, cultures were incubated with 15 µM of P.I. (Sigma–Aldrich, St Louis, MO, USA) and 2 µM of Syto9 (Thermo, Waltham, MA, U.S.A.) for 15 min in the dark. Cells were washed in PBS three times by centrifugation (4000 x g for 5 min). Then cell suspension was smeared onto a glass slide and analyzed on an Axio Imager M2 microscope (Carl Zeiss do Brazil Ltd.a, São Paulo, SP, Brazil). Fluorescence was captured of live/dead cells and differential interference contrast (D.I.C.) images for each field of view from multiple areas for analysis. Quantification was performed using the Knime workflow of different bacteria colonies [82].

### 2.7. Measurement of Reactive Oxygen Species (ROS) Levels 

Intracellular ROS levels were measured in O_3_ treated and not treated bacteria (*A. baumannii* MDR, *E. coli* (ATCC 25922), and *P. aeruginosa* (ATCC 15442). Bacteria suspension in B.H.I. (10^3^ CFU ml^−1^) was cultivated in plates of 24 wells for 10 h in the presence of O_3_. The bacteria were incubated for 30 min in H_2_O_2_ (1% *v*/*v*) as positive controls. After incubation, bacteria were loaded with 20 µM H_2_DCFDA for 45 min. The fluorescence signal, generated by the probe’s oxidation by intracellular ROS, was measured using 485/535 nm excitation/emission wavelengths with a SpectraMax M2 microplate reader (Molecular Devices, CA, USA) [83].

Each experiment was repeated three times for each microorganism treated with O3. The paired *t*-test was applied to compare the statistical significance between the two parts (with and without treatment with O_3_). Differences were determined to be significant if *p* ≤ 0.05.

### 2.8. Scanning Electron Microscopy (SEM)

Morphological changes in the bacteria species were visualized using SEM. For analysis, control cells or O_3_ treatment were fixed for 1 h with 2.5% glutaraldehyde in 0.1 M cacodylate buffer. After fixation, the cells were washed three times in PBS for 5 min, post-fixed for 15 min in 1% osmium tetroxide (OsO4), and rewashed three times in PBS for 5 min. Next, the samples were dehydrated in an ascending series of ethanol (7.5, 15, 30, 50, 70, 90, and 100% ethanol) for 15 min each step, critical point dried with CO_2_, sputter-coated with a 15-nm thick layer of gold and examined in a Jeol JSM 6390 (Tokyo, Japan) scanning electron microscope.

## 3. Results

### 3.1. Monitoring of Ozone Concentration

The monitoring of the O_3_ concentration in the test room (27 °C) showed that the device’s emission ranged from 0.6 ppm to 2.1 ppm, with the average of all measurements around 1.4 ppm.

### 3.2. Ozone Treatment

The culture exposure at different times (10 h and 12 h) with a low-level of gaseous O_3_ did not wholly prevent the growth in vitro of all tested bacterial strains (Figure 1). No reduction in colony count compared to the control group (not treated with O_3_) was observed as statistically significant. For strains *A. baumannii* MDR and *S. aureus* ATCC 6538, the count of bacterial colonies in the control group was higher than those that underwent treatment with O_3_, but with no statistically significant difference. In the other strains of *S. enterica*, ATCC 10708, *P. aeruginosa* ATCC 15442, and *E. coli* ATCC 25922, O_3_ did not affect bacterial proliferation compared to the baseline group.

However, as the colony count does not provide information regarding the bacteria’s metabolic, functional, and proliferative capacity, the cell viability assay in solution was investigated using resazurin as a metabolic indicator. In this assay, after O_3_ treatment, three random colonies were incubated in TSB medium, and after 24 h, the viability of the different species was measured (Figure 2). A significant difference was observed in the viability of *A. baumannii*, *E. coli*, and *P. aeruginosa* treated with O_3_. Ozone treatment mainly reduced the bacterial growth of *E. coli*, leading to an inhibition of about 30%, followed by *P. aeruginosa* (25%) and *A. baumannii* (15%). After the O_3_-treatment of *S. aureus* and *S. enterica* found no differences in the viability.

### 3.3. Effect of Ozone

The use of P.I. as a membrane permeability indicator showed that the bacterial membranes of all strains from ATCC were not affected by ozone. No difference statistic was observed compared to untreated control (Figure 3). Representative fluorescence images comparing untreated and Ozone treated groups also showed no difference in the distribution of PI permeable bacteria (Figure 4).

### 3.4. ROS Analysis 

Since O_3_ diffuses in solution and decomposes to elemental oxygen and free radicals, we measured the oxidative stress produced by ozone treatment using a fluorescent probe. Results demonstrate a significant increase of ROS in the ozone-treated group for the three pathogenic bacteria. Compared to the untreated control, the increase in ROS was higher in *E. coli* (90 ± 14%), followed by *P. aeruginosa* (62.5 ± 19%) and *A. baumanni* (52.6± 5%) (Figure 5). *E. coli* is more sensible to oxidizing agents since H_2_O_2_ leads to an increase of approximately 215% compared to basal ROS production of the control group. *A. baumannii* and *P. aeruginosa* had a similar increase upon H_2_O_2_ treatment, 171 and 173%, respectively (Figure 5).

### 3.5. Scanning Electron Microscopy (SEM.)

Scanning electron microscopy was performed to confirm membrane damage to bacterial species. Morphological analysis showed a striking effect of O_3_ in *A. baumannii*. Treated bacteria showed many protrusions that resemble membrane blebbing. All bacterial controls showed smooth and homogeneous surfaces. Both *E. coli* and *P. aeruginosa* present membrane alterations after O_3_ treatment. The treatment produced in bacterial membrane wrinkle cells with damaged areas with invagination (Figure 6).

## 4. Discussion

Antimicrobial resistance is a growing and worrying concern inside and outside the hospital environment. Although several studies have shown that the bactericidal effect of O_3_ is better with higher concentrations of exposure than lower concentrations [84,85], little is known about its bactericidal effect when used in low concentrations since the results are limited or inconclusive. Therefore, in this study, we have evaluated the antimicrobial efficiency of a low concentration of O_3_ in five bacteria, measuring the cell viability and cytotoxic effect. Four of these bacterial strains are recommended by the Association of Official Analytical Chemists [86] and the National Institute of Quality Control of Brazil (I.N.C.Q.S.; P.O.P. No. 65.3210.007 [87] and one multidrug-resistant (M.D.R.) *A. baumannii*, a bacterium commonly present in patients with severe nosocomial infections. 

The concentration of O_3_ produced by an Ozonator depends on the size of the area, equipment capacity, if there are open doors, or if there are materials that react with O_3_ on the site [87,88]. 

Major regulatory bodies have issued rules and laws that regulate the maximum number of hours allowed for particular gas concentrations in the workplace. The United States Department of Labor’s Occupational Safety and Health Administration (OSHA) allows maximum exposure of 0.1 ppm of O_3_ in 8-h work environments [89]. The Environmental Protection Agency (EPA.), through the Code of Federal Regulations (CRF.) Tile 21, allows maximum exposure of 0.05 ppm of gas [90]. In Brazil, safe dosages of O3 in work environments are indicated by the Ministry of Labor, through the Regulatory Norm ((NR) 15, annex 11) [91], which means a maximum exposure of 0.08 ppm (0.16 mg/m^3^) for working hours of up to 48 h per week. There are factors in the work environment that must be considered, such as ventilation and other elements that act in the destruction of the O_3_ molecule. 

Measurements were carried out to estimate the O_3_ concentration in ambient air generated by the air purifier/sterilizer with O_3_, directly on the output grid of the SANITECH O3 PURI-MU device with the emission ranging from 0.6 to 2.1 ppm. The Eco Sensors Division of K.W.J. Engineering Inc, which is the leading designer of gas detection and O_3_ detection instruments for industrial environments and personal protection, meeting the safety needs of workers in detecting O_3_ and gas since 1992 [92], determines that the O_3_ concentration decreases rapidly as the measuring distance from the generator increases. For example, ozone that reads 10 ppm directly in the output grid rarely exceeds 0.1 ppm at 1 m from the generator [93]. Therefore, according to our results, we can presume that at a 1-m distance, the O_3_ level is approximately 0.006 to 0.021 ppm which is lower than 0.1 ppm, the legally permissible exposure limit set by the Occupational Safety and Health Administration (O.S.H.A.) of safe levels of O_3_ emission in busy environments ensuring its non-toxicity. 

The effectiveness of O_3_ for disinfecting surfaces and rooms has been examined in several previous studies, and many of them have used low O_3_ concentrations because of the toxic nature of the gas and its relatively long half-life [69,94,95,96,97]. Nevertheless, low O_3_ concentrations generally lead to limited or inconclusive results. Despite not obtaining a significant reduction of in vitro bacteria growth, when we investigated its metabolic capacity through resazurin, we found a significant decrease in values for three of the five bacteria studied, showing that even at low concentrations, O_3_ was able to interfere with cell viability. In addition, we found that O_3_ presented the highest inhibitory effect on *E. coli* (30%), followed by *P. aeruginosa* (25%) and *A. baumannii* (15%). Among bacteria, *E. coli* is the most sensitive to O_3_. 

Low doses of O_3_ can effectively control cryptosporidium and *Mycobacterium avium* [98]. A study showed that a single topical application by nebulizing a low amount of O_3_ could completely inhibit the growth of several potentially pathogenic bacterial strains with known resistance to antimicrobial agents [99]. On the contrary, gram-positive cocci (Staphylococcus and Streptococcus) and viruses are more resistant to O_3_. Although, ozone can sterilize both Gram-positive and Gram-negative bacteria [100]. Ozone is an unstable molecule that rapidly decays to O_2_ and releases a single oxygen atom. The single oxygen atom reacts with the cell membrane of the bacteria, attacks the cellular components, interrupts the regular cell activity, and then destroys bacteria [100,101,102]. Another study verified the effectiveness of ozone as a terminal disinfectant by evaluating different microorganisms inoculated in stainless steel squares and incubated at various temperatures and relative humidity for up to 4 h. Contaminated yards were set to be identically compared after exposure to O_3_ (2 ppm/4 h) the survival of these microorganisms. The disclosure of O_3_ to the dirty surfaces resulted in a reduction in microbial viability that varied depending on the type of organism (7.56 to 2.41 log values), suggesting that, if applied after adequate cleaning, ozone can be used as an effective terminal disinfectant [103]. An in vitro study observed that O_3_ effectively reduces concentrations of *A. baumannii*, Clostridium difficile, and methicillin-resistant *S. aureus* (MRSA) in dry and wet samples, suggesting that it can be used as a disinfectant [52]. 

Oxidative stress is defined as a disturbance of the prooxidant/antioxidant balance in favor of prooxidants, leading to potential damage to the cell [104]. Excess pro-oxidants result in oxidative stress, damaging cell components such as proteins, lipids, and DNA. oxidation [105,106]. Oxidative stress induces free radicals with highly reactive elements, such as reactive oxygen species (ROS), which can attack biological molecules and lead to death [107]. The O_3_ is a strong oxidant that generates reactive oxygen species (ROS.) in tissue and causes N.A. damage [108]. In our analysis, the permeability of the bacterial membrane of all strains from ATCC was not significantly affected by ozone, with no difference in the distribution of PI-positive bacteria. However, we observed a significant increase of ROS in the ozone-treated group for *E. coli* (90 ± 14%), followed by *P. aeruginosa* (62.5 ± 19%) and *A. baumanni* (52.6 ± 5%) and certain damage to the membrane observed by SEM. 

It is well known that ozone is poisonous; this is an essential factor in air pollution, particularly affecting children; inhalation can damage the lungs with possible serious consequences [109]. However, in common with many other therapies that induce ROS, the outcome of treatment with ozone, at shallow doses, can be beneficial rather than damaging [110,111,112]. 

Although the microbicide effect of O_3_ is known, its mechanism of action is nameless. A previous study evaluated the bactericidal effect under low O_3_ concentration and found that the negative and positive ions generated during exposure induced oxidative stress, including oxidation of amino acids, cell wall reaction [113], and DNA damage [114], causing cell death. It has been hypnotized that the cell lysis depends on the extent of the reaction [113]. The primary cellular targets for O_3_ are nucleic acids, where damage can range from base lesions to single and double-strand breaks [115]. Lesions can lead to more or less compromising point mutations, whereas massive DNA breakage is lethal if not repaired [116,117,118]. Many other studies provide evidence that the cell envelope is affected during ozonation, probably even before severe DNA damage occurs [119,120,121]. 

The effectiveness of O_3_ as a disinfectant varies significantly between different types of bacteria, even at the strain level as shown previously [115,122] and by our analysis. However, its effect depends on several intrinsic factors, such as growth stage, cell envelope, the efficiency of repair mechanisms, and the type of viability indicator used [123,124,125]. Moreover, other extrinsic factors, such as concentration and type of dissolved organic material or the presence of flakes or particles, may reduce O_3_ stability or may protect microorganisms from its effects, thereby decreasing the disinfection efficiency [126,127,128,129]. Nevertheless, ozone gas has been successfully applied to disinfection viruses on surfaces and aerosols [130,131].

## 5. Conclusions

This work shows that low ozone doses did not inhibit bacterial growth but may interfere with the cell viability of the three bacterial strains studied. There was no damage to bacterial membranes, but measurement of oxidative stress showed a significant increase in intracellular levels of reactive oxygen species (ROS) that damage lipids, proteins, and DNA. Furthermore, significant morphological changes, such as protrusions resembling membrane blisters and/or invaginations, were observed in *E. coli* and *P. aeruginosa* after treatment with O_3_.

These results are promising and encourage further investigations into using gaseous ozone at low concentrations as a disinfectant or antiseptic, evaluating its bactericidal effect, as it can reduce the transmission of microorganisms and is essential in the maintenance and/or disinfection of health environments.

## Figures and Tables

**Figure 1 microorganisms-10-00040-f001:**
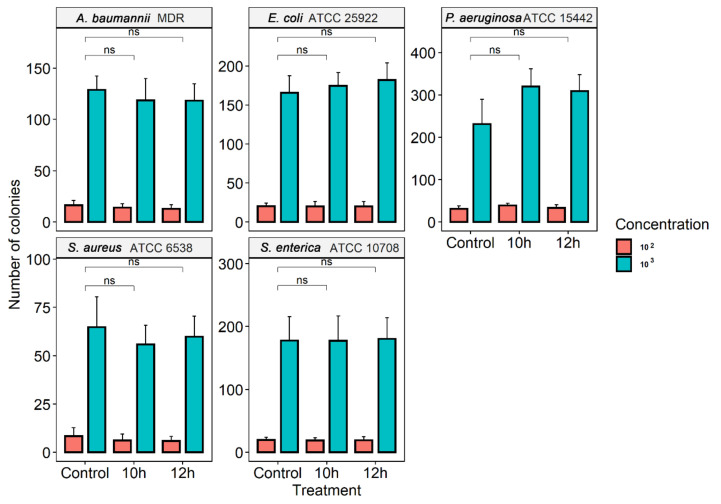
Counting the number of colonies in different species of bacteria (*A. baumannii* MDR, *S. aureus* (ATCC 6538), *S. enterica* (ATCC 10708), *E. coli* (ATCC 25922), and *P. aeruginosa* (ATCC 15442). Quantification of the number of colonies was performed in the control group (without treatment) and in bacterial suspensions (10^3^ and 10^2^ CFU/mL) after exposure to O_3_ for 10 h and 12 h; ns: statistically not significant using the *t*-test.

**Figure 2 microorganisms-10-00040-f002:**
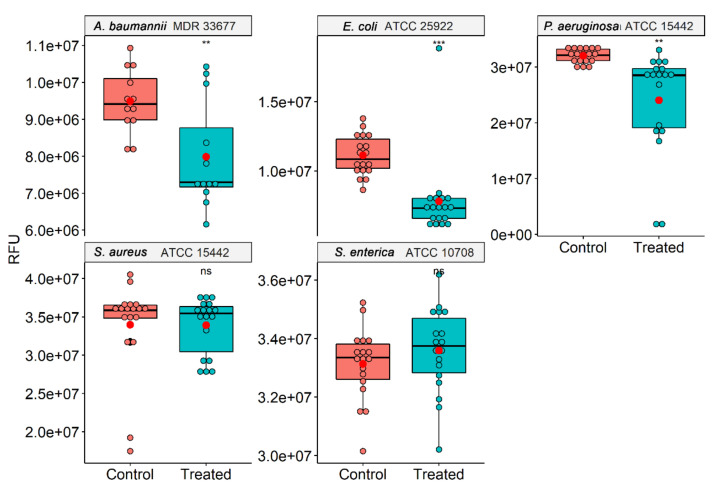
Viability of different bacteria (*A. baumannii*, *E. coli*, *P. aeruginosa, S. aureus*, and *S. enterica*) after exposure to ozone for 10 h using resazurin as a metabolic indicator. Viable bacteria convert resazurin into fluorescent resorufin, comparison of O_3_ exposure of different colonies was visualized in a boxplot. Red dots represent the mean of the relative fluorescence from each group. Statistically significant using *t*-test (** *p* < 0.01; *** *p* < 0.001).

**Figure 3 microorganisms-10-00040-f003:**
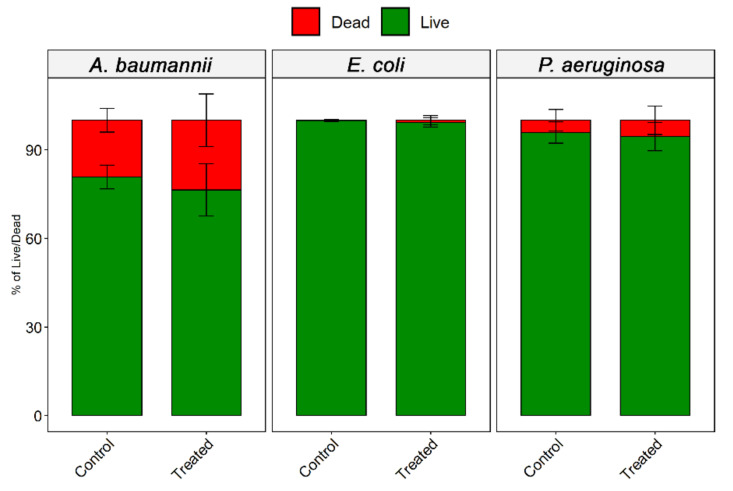
Quantification of live/dead bacteria after exposure to O_3_ for 10 h. Percentage of live and dead cells quantified from different microscopic images.

**Figure 4 microorganisms-10-00040-f004:**
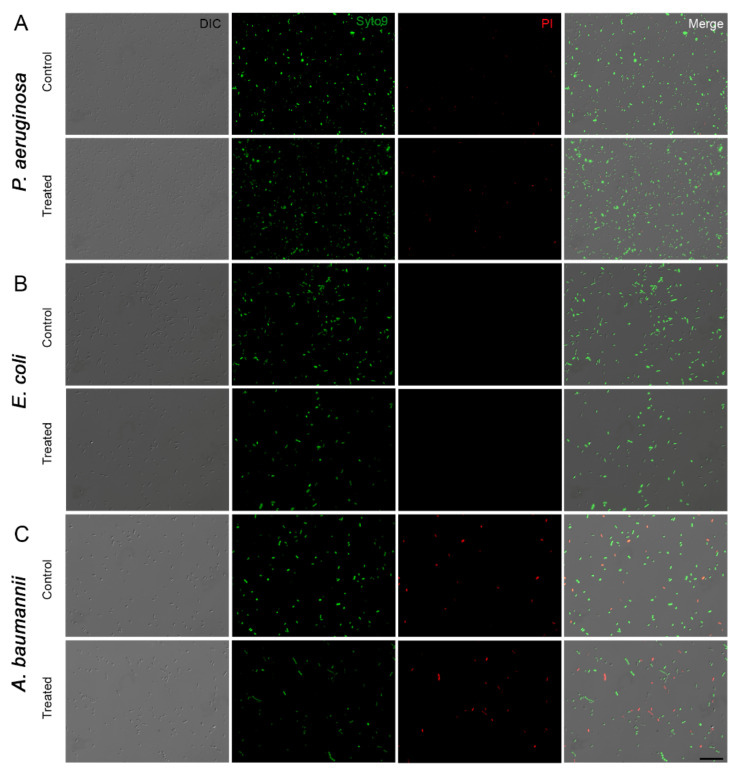
Fluorescence images of *P. aeruginosa* (**A**), *E. coli* (**B**), and *A. baumannii* (**C**) were exposed to O_3_ for 10 h and stained with Syto9 (green) and propidium iodide (PI; red). Live bacteria were evidenced using Syto9, while membrane-permeable dead bacteria were stained with PI DI.C: differential interference contrast. Bar = 20 μm.

**Figure 5 microorganisms-10-00040-f005:**
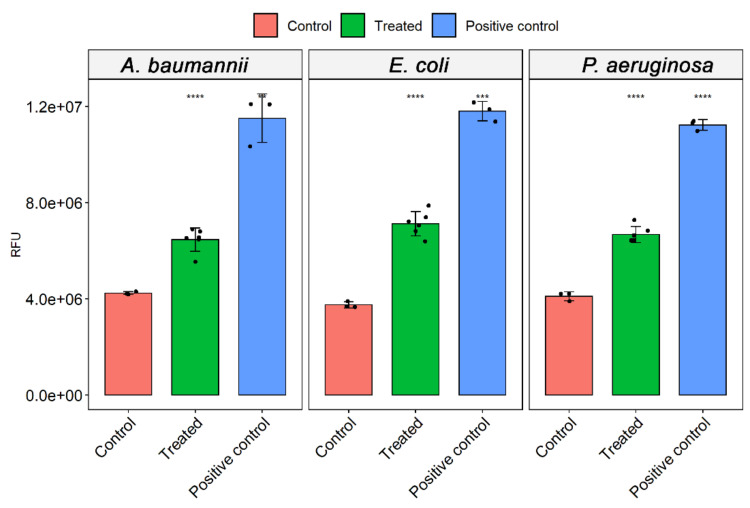
Measurement of reactive oxygen species (ROS.) induced by O_3_ on different bacteria after incubation during 10 h. Hydrogen peroxide (1%) was used as a positive control. The data represent the mean and standard deviation of triplicates using different colonies. *t*-test (***) *p* < 0.01 and (****) *p* < 0.001.

**Figure 6 microorganisms-10-00040-f006:**
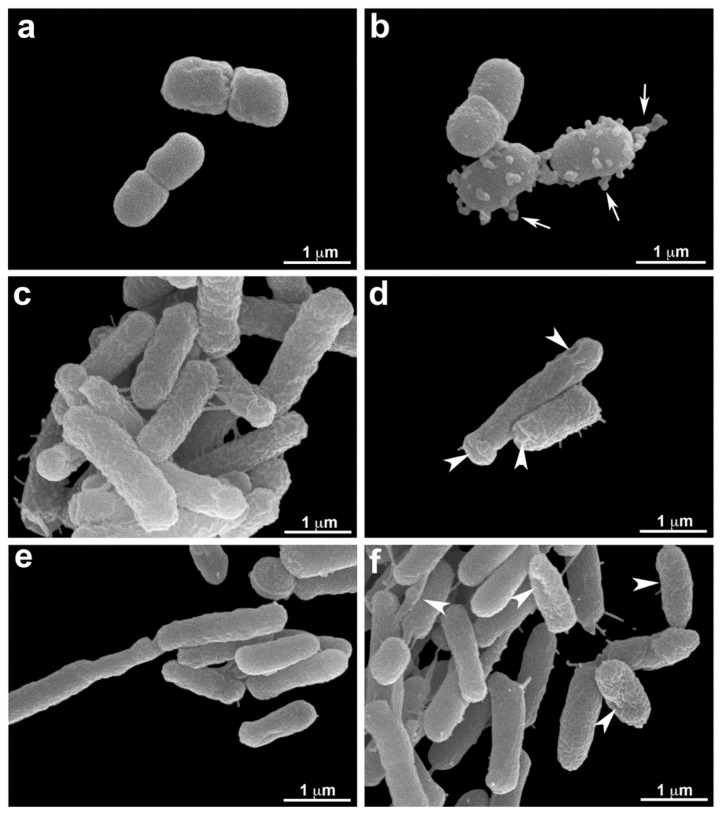
Morphological analysis of O_3_ treatment by electron microscopy. *A. baumannii* (**a**,**b**), *E. coli* (**c**,**d**), and *P. aeruginosa* (**e**,**f**) are seen without (**a**,**c**,**e**) and under O_3_ treatment (**b**,**d**,**f**). After treatment, there are membrane protrusions (arrows) are seen in *A. baumannii* (**b**). Note that control cells exhibit a homogeneous surface (**a**). Both *E. coli* and *P. aeruginosa* present surface alterations after treatment (**d**,**f**). Observing more wrinkle cells is possible than control cells (**c**,**e**), and some damage is verified (arrowhead).

## Data Availability

The data presented in this study are available on request from the corresponding author.

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
