# Peer review of "Detrimental Effect of Ozone on Pathogenic Bacteria"

_microorganisms, 2021, doi:10.3390/microorganisms10010040_

Round 1

Reviewer 1 Report

Antibiotic resistance is a big problem and new or combinations of old methods are needed to fight it. Ozonation is one of the methods that could be applied in hospital settings, but not only in hospital settings. There are a lot of errors in the manuscript and the names of the bacteria should be written in italic in the text, but also in the charts. In vitro terms should also be written in italic. In the introduction, too much has been written about the issue of resistance, and little about the actual topic being examined in the manuscript. it should be reorganized.
My main complaint is the use of nutrient medium as a medium on which bacteria are present during ozone exposure. Why did you choose a nutrient medium and not one of the surfaces such as metal, plastic, textile or ceramic? During 10 hours of exposure, was the ozone concentration constant? Namely, does ozone have a half-life?
It is not clear to me whether the experiments in which you examined oxidative stress or viability were performed on agar or broth? If they are in the broth how do you know how bad the ozonation bacteria were in general? Namely, it is not so easy to push gas into a liquid.

Author Response

  • Antibiotic resistance is a big problem, and new or combinations of old methods are needed to fight it. Ozonation is one of the methods that could be applied in hospital settings, but not only in hospital settings. There are a lot of errors in the manuscript, and the names of the bacteria should be written in italic in the text and the charts. In vitro terms should also be written in italic.

      R: The names of the bacteria were rechecked and adjusted in the text and charts.

  • In the Introduction, too much has been written about the issue of resistance and little about the actual topic being examined in the manuscript. It should be reorganized.

      R: We inserted this paragraph in the Introduction. Lines 120-127, lines 134-135.

  • My main complaint is using the nutrient medium as a medium on which bacteria are present during ozone exposure. Why did you choose a nutrient medium and not one of the surfaces such as metal, plastic, textile, or ceramic? During 10 hours of exposure, was the ozone concentration constant? Namely, does ozone have a half-life?

      R: This experiment was a preliminary test to evaluate the effect of low concentrations of ozone initially on four bacterial strains recommended by the Association of Official Analytical Chemists and the National Institute of Quality Control in Brazil, used in assays with disinfectants and one multidrug-resistant (MDR) A. baumannii, a bacterium commonly present in patients with severe nosocomial infections. Later, we intend to test other surfaces that mimic both the home and hospital environment.

      During 10 hours of exposure, the ozone concentration ranged from 0.6 ppm to 2.1 ppm, with the average of all measurements around 1.4 ppm.

      The half-life of ozone depends on temperature, airflow, and relative humidity. Under standard conditions (21°C, 1 bar), the half-life of ozone gas is 20 minutes.

  • It is not clear whether the experiments in which you examined oxidative stress or viability were performed on agar or broth? If they are in the broth, how do you know how harmful the ozonation bacteria were in general? Namely, it is not so easy to push gas into a liquid.

      R: Viability tests and reactive oxygen species measures were performed in broth. Can find the information in the material and methods section. We measured the ozone concentration produced by the equipment, and bacteria were analyzed by various techniques that demonstrated that species were affected differently by ozone treatment. Ozone is a gas that diffuses well in liquid and is traditionally used for drinking and wastewater treatment. Please see the articles below:       Von Gunten U. Ozonation of drinking water: part I. Oxidation kinetics and product formation. Water Res. 2003 Apr;37(7):1443-67. doi: 10.1016/S0043-1354(02)00457-8. PMID: 12600374.

Reviewer 2 Report

In the era of multidrug resistance, effective solutions against MDR pathogens are highly anticipated therefore, the topic is of particular interest, even if the results are modest but encouraging. The study is well designed and made which is why we congratulate the authors. Our minor suggestions are as follows:

  1. Names of the pathogens must be italic:  r. 110, r. 249-250
  2. it is difficult to understand the value of temperature during exposure to ozone
  3. in the Introduction could be add data about colonisation of patients with MDR pathogens such as Recent Advances in Investigation, Prevention, and Management of Healthcare-Associated Infections (HAIs): Resistant Multidrug Strain Colonization and Its Risk Factors in an Intensive Care Unit of a University Hospital, by  Micle et al.
  4. in the Introduction could be add data about antibiotic consumption in Europe such as Antibiotic consumption and microbiological epidemiology in surgery departments: results from a single study center, by Vesa et al. 

Author Response

In the era of multidrug resistance, effective solutions against MDR pathogens are highly anticipated; therefore, the topic is of particular interest, even if the results are modest but encouraging. The study is well designed and made, so we congratulate the authors.

R: Thank you.

Our minor suggestions are as follows:

Names of the pathogens must be italic:  r. 110, r. 249-250

R: The names of the bacteria were rechecked and adjusted in the text and charts.

It isn't easy to understand the value of temperature during exposure to ozone.

R: The temperature remained constant at 27oC during the entire experiment of bacteria exposure to ozone.

The Introduction could add data about the colonization of patients with MDR pathogens such as Recent Advances in Investigation, Prevention, and Management of Healthcare-Associated Infections (HAIs): Resistant Multidrug Strain Colonization and Its Risk Factors in an Intensive Care Unit of a University Hospital, by Micle et al.

R: The paragraph was introduced in the Introduction (lines 95-104).

The Introduction could add data about antibiotic consumption in Europe, such as Antibiotic consumption and microbiological epidemiology in surgery departments: results from a single study center, by Vesa et al.

R: The paragraph was inserted in the Introduction (lines 76-84).

Round 2

Reviewer 1 Report

The authors have accepted all suggestions and I have no further comments.